# CQ-CNN: A lightweight hybrid classical–quantum convolutional neural network for Alzheimer's disease detection using 3D structural brain MRI

**Mominul Islam** [1,4,5], **Mohammad Junayed Hasan** [2,3,4,5], **M.R.C. Mahdy** [1,5]*

**1** Department of Electrical and Computer Engineering, North South University, Dhaka, Bangladesh, **2** Department of Computer Science, Johns Hopkins University, Baltimore, Maryland, United States of America, **3** Department of Computational Pathology and AI - Allied Health, Mayo Clinic, Rochester, Minnesota, United States of America, **4** Mahdy Research Academy, Dhaka, Bangladesh, **5** NSU Center of Quantum Computing, Plot, Block B, Bashundhara R/A, Dhaka, Bangladesh

* mahdy.chowdhury@northsouth.edu

**Data availability statement:** The datasets used in this research are publicly accessible at the

## Abstract

The automatic detection of Alzheimer's disease (AD) using 3D volumetric MRI data is a complex, multi-domain challenge that has traditionally been addressed by training classical convolutional neural networks (CNNs). With the rise of quantum computing and its potential to replace classical systems in the future, there is a growing need to: *(i)* develop automated systems for AD detection that run on quantum computers, *(ii)* explore the capabilities of current-generation classical-quantum architectures, and *(iii)* identify their potential limitations and advantages. To reduce the complexity of multi-domain expertise while addressing the emerging demands of quantum-based automated systems, our contribution in this paper is twofold. First, we introduce a simple preprocessing framework that converts 3D MRI volumetric data into 2D slices. Second, we propose CQ-CNN, a parameterized quantum circuit (PQC)-based lightweight hybrid classical-quantum convolutional neural network that leverages the computational capabilities of both classical and quantum systems. Our experiments on the OASIS-2 dataset reveal a significant limitation in current hybrid classical-quantum architectures, as they face difficulties converging when class images are highly similar, such as between moderate dementia and non-dementia classes of AD, which leads to gradient failure and optimization stagnation. However, when convergence is achieved, the quantum model demonstrates a promising quantum advantage by attaining state-of-the-art accuracy with far fewer parameters than classical models. For instance, our $\beta_8$-3-qubit model achieves 97.5% accuracy using only 13.7K parameters (0.05 MB), which is 5.67% higher than a classical model with the same parameter count. Nevertheless, our results highlight the need for improved quantum optimization methods to support the practical deployment of hybrid classical-quantum models in AD detection and related medical imaging tasks.

following links: http://preprocessed-connectomes-project.org/NFB_skullstripped and https://sites.wustl.edu/oasisbrains/home/oasis-2. The models and code are available at: https://github.com/mominul-ssv/alz-cq-cnn.

**Funding:** The author(s) received no specific funding for this work.

**Competing interests:** The authors have declared that no competing interests exist.

## Introduction

Alzheimer's disease (AD) is a progressive neurodegenerative disorder that primarily affects the elderly, leading to cognitive decline and memory loss [1,2]. Currently, there is no proven cure or way to reverse the progression of AD, and it is primarily managed through supportive care provided by healthcare professionals [3–5]. AD pathology is characterized by the accumulation of abnormal proteins, such as amyloid $\beta$ ($A\beta$) and tau ($\tau$), in the brain. These proteins interfere with communication between brain cells, altering their function and ultimately leading to cell death [6,7]. As brain cells die, key areas involved in cognition, particularly the hippocampus, begin to shrink and its degeneration is closely related to the memory loss characteristic of AD [8,9].

Structural changes in the brain are commonly detected using magnetic resonance imaging (MRI) [10,11]. However, manually interpreting MRI scans for tasks such as AD detection is time-consuming and requires expert knowledge [12], which has driven the development of automatic diagnostic systems. A common approach involves training machine learning (ML) models on sample images, such as MRI scans from past patients. Since MRI datasets are typically stored in the NIfTI format, which contains 3D volumetric data, these data must be converted into 2D slices before they can be used to train ML models such as convolutional neural networks (CNNs) [13,14]. Software tools such as FSL, FreeSurfer, ANTs, and ANTsX are commonly used for this conversion; however, they require considerable domain-specific expertise and often have steep learning curves [15,16], necessitating the development of a framework that simplifies the process.

Training classical CNNs with MRI images for early detection of AD has been extensively studied. Yagis *et al.* (2020) proposed the use of 3D CNNs for AD diagnosis using structural MRI data, while Cheng *et al.* (2017) introduced a multi-domain transfer learning approach to improve early-stage AD detection [17,18]. Guan *et al.* (2021) further advanced this field by developing a multi-instance distillation scheme that transfers knowledge from multi-modal data into an MRI-based network [19]. Although these methods show significant progress in automated AD detection with CNNs, they are all designed for classical computing systems. With the rapid development of quantum computing, which has the potential to provide exponential speedups for certain computational tasks compared to classical computers [20], it is becoming increasingly important to design automated AD detection systems that run on quantum computers and can leverage quantum computational capabilities.

Quantum machine learning (QML) has emerged with the goal of translating classical machine learning (CML) concepts into frameworks suitable for execution on quantum hardware [21]. In CML, training is performed on classical computers, which rely on voltage or charge to represent bits that correspond only to two states: 0 and 1. Logical operations are executed using gates such as AND, OR, and NOT, grounded in Boolean algebra and classical physics. In contrast, in QML, training is performed on quantum computers, which use quantum bits (qubits) that exploit quantum properties such as electron spin [22]. Qubits can represent not only classical binary states (0 and 1) but also superposition states, where a qubit simultaneously encodes multiple possibilities, including negative and complex values [23]. This ability to represent a broader range of data, along with quantum phenomena such as entanglement and parallelism, enables quantum computers to perform certain computations with significantly greater efficiency [24].

Several recent studies have explored the integration of classical and quantum computation through hybrid architectures that aim to leverage the computational advantages of both paradigms. For example, Mari *et al.* (2020) introduced a hybrid transfer learning framework that incorporates a pre-trained classical neural network with a variational quantum circuit

(VQC), tested on real quantum hardware [25]. Konar *et al.* (2023) proposed a shallow hybrid classical–quantum spiking neural network (SQNN) by combining VQCs with spiking neurons. Their model demonstrated enhanced noise-robust image classification compared to traditional spiking neural networks, recurrent quantum neural networks, and well-known convolutional architectures such as AlexNet and ResNet-18 [26]. Senokosov *et al.* (2024) introduced two hybrid quantum neural network architectures. One used parallel quantum circuits, and the other incorporated a quanvolutional layer [27]. Both models achieved high accuracy on benchmark datasets such as MNIST and CIFAR-10 while using significantly fewer parameters. Complementing these efforts, Hasan *et al.* (2023) proposed a PQC-based model that achieved classification accuracy comparable to classical models on similar datasets [28].

A consistent theme across these studies is the emphasis on hybrid models that aim to utilize the complementary strengths of classical and quantum computing. Despite promising outcomes, most of these models have been evaluated only on benchmark datasets such as MNIST and CIFAR-10, where inter-class differences are visually distinct. This raises a critical research question: how effective are such hybrid models in complex, fine-grained classification tasks, such as detecting AD from 2D MRI slices extracted from 3D volumetric brain scans, where inter-class variations are subtle and often imperceptible? In this paper, we aim to answer this question. We begin by developing a simple framework that converts 3D MRI volumetric data into 2D slices. We then introduce CQ-CNN, a parameterized quantum circuit (PQC)-based hybrid classical–quantum convolutional neural network. Next, we preprocess the 3D MRI data from the OASIS-2 AD classification dataset using our 3D-to-2D conversion framework and train our models for the binary classification task. Finally, we evaluate the performance of our model, identify potential signs of quantum advantage, explore existing bottlenecks, and investigate the underlying factors contributing to any observed limitations.

The remainder of this paper is organized as follows. The *Methods* section details the 3D-to-2D data conversion framework and provides an overview of the CQ-CNN model architecture. The *Experiments* section describes the dataset used in this study, the preprocessing steps, and the training of the CQ-CNN models. It also outlines the training configurations and the progression of the training process. The *Results* section presents the performance of the CQ-CNN models and analyzes anomalies encountered during training, along with potential explanations. It also includes comparative performance analyses with classical state-of-the-art (SOTA) models. The *Discussion and limitations* section interprets the key findings, discusses possible solutions to the observed issues, suggests directions for future work, and addresses the limitations of the study. Finally, the *Conclusion* section summarizes the main contributions of this work.

## Methods

### 3D to 2D slice conversion

To convert 3D MRI data into 2D slices, consider the 3D volume $V \in \mathbb{R}^3$, where each point represents a voxel in the scanned region. The data can be visualized from three primary anatomical views: the axial plane (where the $\mathbb{R}^{xy}$ plane moves along the z-axis), the coronal plane (where the $\mathbb{R}^{yz}$ plane moves along the x-axis), and the sagittal plane (where the $\mathbb{R}^{zx}$ plane moves along the y-axis), as shown in Fig 1(a) and 1(b). Let $n$ represent the number of slices to be extracted from each anatomical view, and let $m$ denote the total number of slices available in that view. The interval between consecutive slices is denoted by $i$, which determines the spacing between each slice. To calculate the necessary interval $i$ for extracting $n$

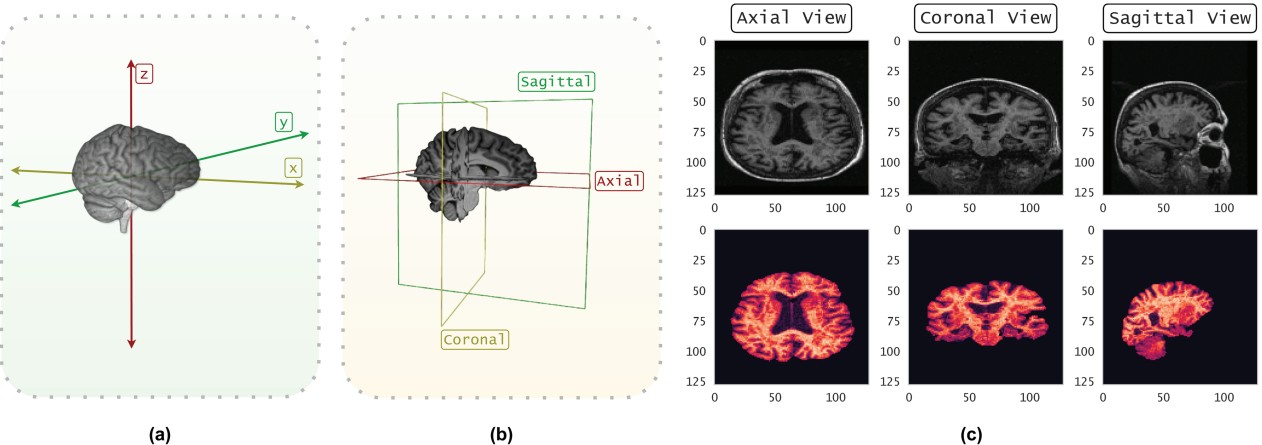

**Fig 1. Subfigure (a) shows the 3D MRI volume represented as voxels in a three-dimensional coordinate system; (b) presents example 2D slices from the axial, coronal, and sagittal planes; and (c) displays corresponding MRI images from these planes with non-skull-stripped images in the top row and skull-stripped images in the bottom row.**

slices from $m$ total slices, the following equation is used:

$$i = \left\lfloor \frac{m}{n} \right\rfloor \tag{1}$$

where $\lfloor \cdot \rfloor$ denotes the floor function, which rounds the slice intervals down to the nearest integer. However, in MRI data, the first and last few slices often do not contain meaningful voxels due to the absence of relevant tissue. Therefore, these slices are excluded after determining the interval $i$. The total number of valid slices is reduced by $k_1$ slices from the beginning and $k_2$ slices from the end, as illustrated in Fig 2. The final number of slices to be

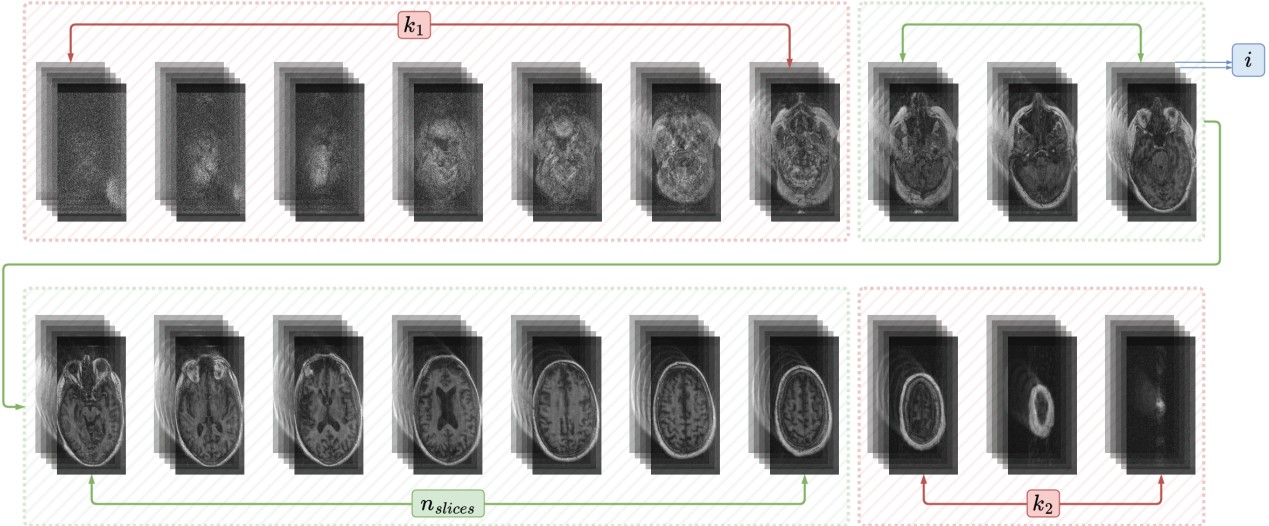

**Fig 2. Visualization of the 3D-to-2D slice extraction strategy from volumetric MRI data (axial view).** The slice interval $i$, calculated using Eq 1, defines the spacing between the selected slices. To exclude boundary regions that primarily contain empty space or non-brain tissue, the first $k_1$ and the last $k_2$ slices are discarded. The remaining central $n_{\text{slices}}$ slices, calculated using Eq 2, represent the feature-rich region.

selected is then:

$$n_{\text{slices}} = \left\lceil \frac{m}{i} \right\rceil - (k_1 + k_2) \qquad (2)$$

where $\lceil \cdot \rceil$ denotes the ceiling function, which rounds the slice number up to the nearest integer. We use Eq 2 to ensure that the final set of slices extracted from the volume $V$ is evenly distributed along the anatomical planes $\mathbb{R}^{xy}$, $\mathbb{R}^{yz}$, or $\mathbb{R}^{zx}$, preserving the important structural information from the 3D volume while eliminating irrelevant regions at the edges.

**Algorithm 1 3D-to-2D MRI slice extraction.**

```
Input: 3D volume V ∈ ℝ^{X×Y×Z}, target slices n, total slices m,
       edge exclusions k₁,k₂, plane P ∈ {axial,coronal,sagittal}
Output: Evenly spaced 2D slices S
 1: Initialize S ← ∅
 2: Compute the interval i (Eq 1)
 3: Compute the number of valid slices to extract, n_slices (Eq 2)
 4: for j = 0 to n_slices − 1 do
 5:   Calculate slice index s ← k₁ + j · i    6:    if s < m − k₂ then
 7:      Extract slice V_s from V along P
 8:      Append V_s to S
 9:   end if
10: end for
11: return S
```

We implement this slice selection strategy as the core component of our 3D-to-2D data transformation framework, and the complete pseudocode is presented in Algorithm 1.

## Classical vs. quantum neural network

A classical neural network is a computational model inspired by the structure of the human brain. It consists of interconnected layers of artificial neurons (or perceptrons), where each neuron processes input data through an activation function (such as ReLU, sigmoid, or tanh) and transmits the output to subsequent layers. A typical classical neural network includes three types of layers: an input layer, one or more hidden layers, and an output layer (as illustrated in Fig 3(a)).

In contrast, a quantum neural network (QNN) is a hybrid machine learning model that combines classical computation with quantum processing. At its core lies a PQC, which consists of three key components: *(i)* data encoding, *(ii)* an ansatz (a circuit with trainable quantum gates), and *(iii)* quantum measurement, as shown in Fig 3(b). These operations are executed on a quantum computer. The parameters within the PQC are then optimized using a classical optimization algorithm, forming a feedback loop between the quantum and classical systems.

## Parameterized quantum circuit

**Data encoding.** The first component of a PQC is data encoding. Several common encoding techniques, such as angle encoding, amplitude encoding, and basis encoding, are used to map classical data onto quantum states. Angle encoding represents classical data as parameters for rotation gates (such as $R_x$ and $R_y$), where the input data directly determine the angles of these gates. Amplitude encoding maps classical data to the amplitudes of quantum states, where the data is represented as a superposition of basis states with complex amplitudes.

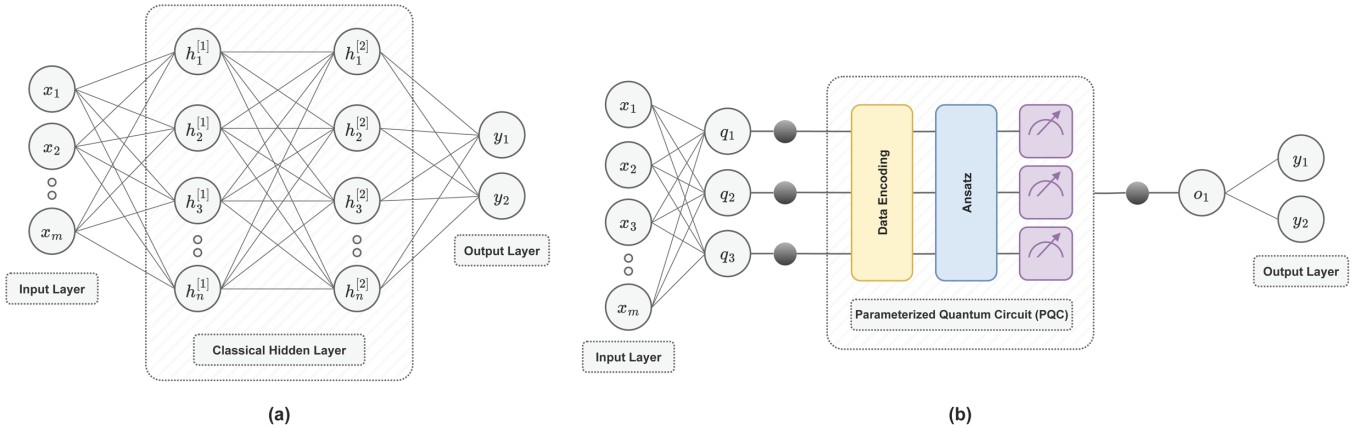

**Fig 3. Schematic depiction of a classical neural network (a) and a quantum neural network (b) for binary classification.** In subfigure (a), $x_1, x_2, \ldots, x_m$ denote the $m$ input neurons representing the input features. The hidden layer consists of $n$ neurons represented as $h_1^{[1]}, h_2^{[1]}, \ldots, h_n^{[1]}$, where the superscript [1] indicates the first hidden layer, and the subscript identifies the specific neuron within that layer (e.g., $h_1^{[1]}$ is the first neuron in the first hidden layer). The output layer neurons, representing the predicted probabilities for each class given the input features, are denoted by $y_1$ and $y_2$. In subfigure (b), the input and output layers are similar to those in subfigure (a). However, the classical hidden layers are replaced by a 3-qubit PQC. The classical features are first reduced to match the number of qubits, represented as $q_1, q_2, q_3$, with three black dots indicating the qubits. These features are then encoded into quantum states through data encoding. A parameterized ansatz is applied to capture complex relationships using quantum operations. Afterward, quantum measurements are performed, and the PQC outputs a classical probability. This probability passes through an intermediate linear layer, denoted as $o_1$. Finally, $o_1$ is mapped to the output probability using Eq 9.

Basis encoding, on the other hand, assigns classical data directly to specific quantum basis states (such as $|0\rangle$, $|1\rangle$, etc.), where each classical value corresponds to a particular state in the computational basis.

ZZFeatureMap is a relatively new encoding technique that extends traditional angle encoding by introducing entanglement between qubits, and is used in our PQC [29]. The process begins with state preparation, where each qubit is initialized in a Hadamard ($H$) state, creating a superposition of $|0\rangle$ and $|1\rangle$. The feature map then applies parameterized gates $P(2 \cdot x[i])$ to each qubit, where $x[i]$ represents the classical data (as shown in the initial phase of the PQC in Fig 4(a) and Fig 4(b)). These gates adjust the phase of each qubit based on the corresponding classical input values.

Next, entanglement is introduced through controlled-Z (CZ) gates, which create correlations between pairs of qubits. This entanglement spreads the classical data across multiple qubits, allowing the quantum system to represent complex correlations that are challenging for classical models to capture. Mathematically, the encoding process using the ZZFeatureMap for a $N$-qubit system can be expressed as:

$$|\psi(\mathbf{x})\rangle = H^{\otimes N} \cdot \prod_{i=1}^{N} P(2 \cdot x[i]) \cdot \prod_{i<j} CZ(i,j) \tag{3}$$

where $H^{\otimes N}$ represents the Hadamard operation applied to each qubit, $P(2 \cdot x[i])$ is the parameterized gate acting on each qubit, and $CZ(i,j)$ is the controlled-Z gate applied between qubits $i$ and $j$, creating entanglement. This results in an entangled state $|\psi(\mathbf{x})\rangle$, which encodes the classical data into a quantum state.

**Ansatz.** Following data encoding, the output quantum state $|\psi(\mathbf{x})\rangle$ is passed as input to the ansatz. The ansatz applies a sequence of trainable quantum gates to change the encoded

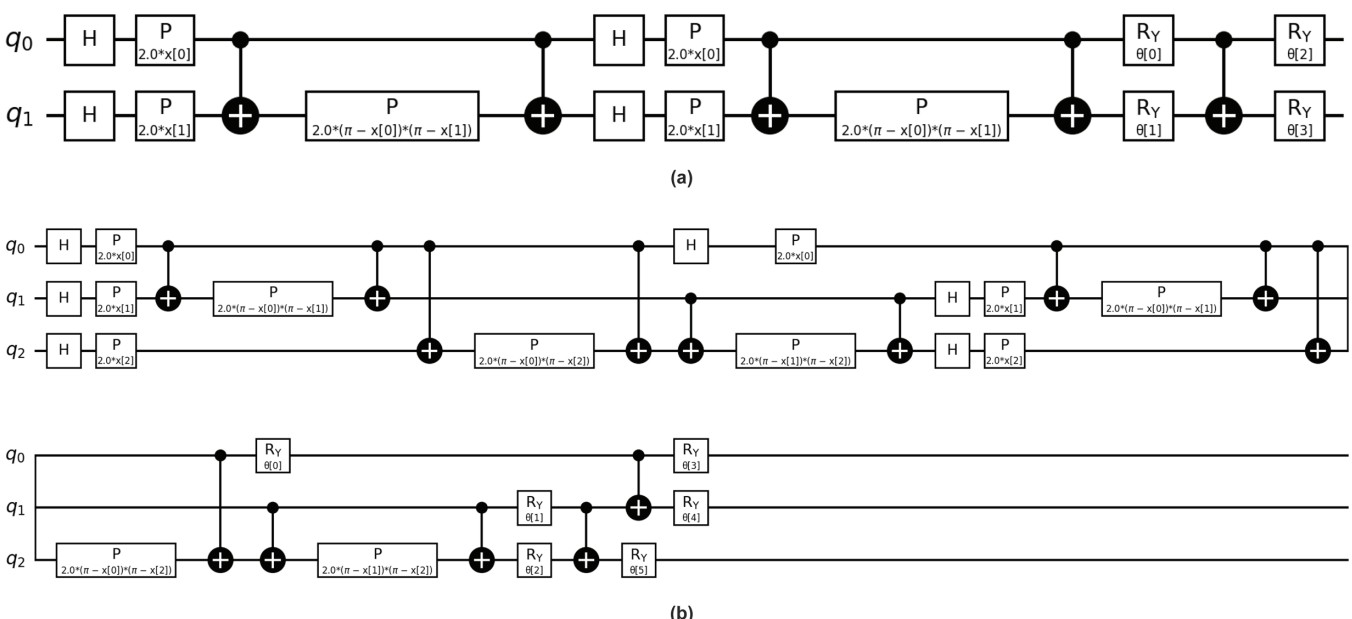

**Fig 4. The schematic depicts our PQC using ZZFeatureMap encoding, with subfigure (a) showing a 2-qubit circuit and subfigure (b) showing a 3-qubit circuit.** Each qubit is initialized with a Hadamard gate $H$, followed by phase rotations $P(2 \cdot x[i])$ to encode classical data into a quantum state. Entanglement is then introduced through controlled-Z (CZ) gates, which create correlations between qubits by applying phase shifts based on their classical values. A phase rotation $P(2.0(\pi - x[i])(\pi - x[i]))$ is applied to introduce further phase shifts based on the classical values. The ansatz circuit applies trainable single-qubit rotations $R_y(\theta_i)$ to further refine the quantum state.

quantum state, allowing it to learn patterns for making predictions. In ZZFeatureMap encoding, the ansatz uses parameterized rotation gates $R_y(\theta_i)$ (depicted at the end of the PQC in Fig 4(a) and Fig 4(b)), where $\theta_i$ represents a trainable parameter for the $i$-th qubit. For an $N$-qubit system, the ansatz is formulated as:

$$|\psi(\theta)\rangle = U(\theta) \cdot |\psi(x)\rangle = \prod_{i=1}^{N} R_y(\theta_i) \cdot |\psi(x)\rangle \tag{4}$$

where $U(\theta)$ represents the parameterized ansatz circuit, and the product notation indicates the sequential application of rotation gates to all $N$ qubits.

**Quantum measurement.** Once the ansatz circuit has transformed the quantum state, the next step is quantum measurement. Measurement collapses the quantum state to one of the eigenstates of the measurement operator, which in the case of PQCs is the Pauli-Z operator $\sigma_z$, representing a computational basis measurement. The measurement results give classical probabilities that can be used to compute the output of the quantum neural network. The probability $p_i$ of obtaining a specific measurement outcome $i$ is given by:

$$p_i = |\langle i|\psi(\theta)\rangle|^2 \tag{5}$$

where $|i\rangle$ represents the $i$-th eigenstate, and $|\psi(\theta)\rangle$ is the quantum state after the ansatz transformation. The expected value of the measurement outcome $M$ can be computed as:

$$\langle M \rangle = \sum_i l_i \cdot p_i = \sum_i l_i |\langle i|\psi(\theta)\rangle|^2 \tag{6}$$

where $l_i$ is the eigenvalue associated with the eigenstate $|i\rangle$, typically +1 or −1 for Pauli-Z measurements.

## Parameter optimization

The parameters $\theta = (\theta_0, \theta_1, \theta_2, \ldots, \theta_m)$ of the ansatz circuit are optimized classically to minimize a loss function $L(\theta)$, which is based on the measured outcomes of the quantum circuit. The loss function used in our QNN for classification tasks is the cross-entropy loss:

$$L(\theta) = -\frac{1}{N} \sum_{j=1}^{N} \sum_{c=1}^{C} y_{jc} \log(p_i = c) \tag{7}$$

where $N$ is the number of samples, $C$ is the number of classes, $y_{jc}$ is the true label, and $p_i = c$ is the probability of measuring the eigenstate corresponding to class $c$.

The classical optimization algorithm, known as gradient descent, is used to update the parameters of the ansatz circuit:

$$\theta_i^{(k+1)} = \theta_i^{(k)} - \eta \cdot \frac{\partial L(\theta)}{\partial \theta_i} \tag{8}$$

where $\eta$ is the learning rate, and the gradient $\frac{\partial L(\theta)}{\partial \theta_i}$ is computed using the parameter-shift rule on the quantum device.

The output of the PQC, $o_1$, is a classical probability value, which is then mapped to the output probability, $\gamma$, using the following equation:

$$\gamma = \text{concatenation}((o_1, 1 - o_1), -1) \tag{9}$$

where the concatenation operation combines $o_1$ with $1 - o_1$ to form the final output vector $\gamma$ (as shown in the output of Fig 5).

## Quantum convolutional neural network

In tasks like image classification, such as detecting AD from MRI images, convolutional neural networks (CNNs) are commonly used. Unlike traditional neural networks, CNNs use specialized layers called convolutional filters to process input data and detect local features such as edges, textures, and shapes. These features are then passed through activation functions and processed by pooling layers, which reduce the spatial dimensions of the feature maps while retaining the most important information. The features are then flattened into a one-dimensional vector and fed into a fully connected layer, which generates the output. For classification tasks, this output is usually passed through a softmax function, which converts the raw output into a probabilistic distribution, where each class is assigned a probability between 0 and 1.

A classical CNN can be transformed into a hybrid classical-quantum convolutional neural network (CQ-CNN) by incorporating a PQC after the flattened one-dimensional vector. To ensure compatibility, we reduce the number of neurons in the fully connected layer so that its connections match the number of qubits in the PQC. In our CQ-CNN architecture, we also replace the softmax layer with Eq 9 to generate the final output probabilities. In the CQ-CNN, as illustrated in Fig 5, the convolutional filters first extract local features from the input MRI slice, which are then processed through ReLU activation functions and max-pooling layers. The resulting feature maps are flattened into a one-dimensional vector and passed through a

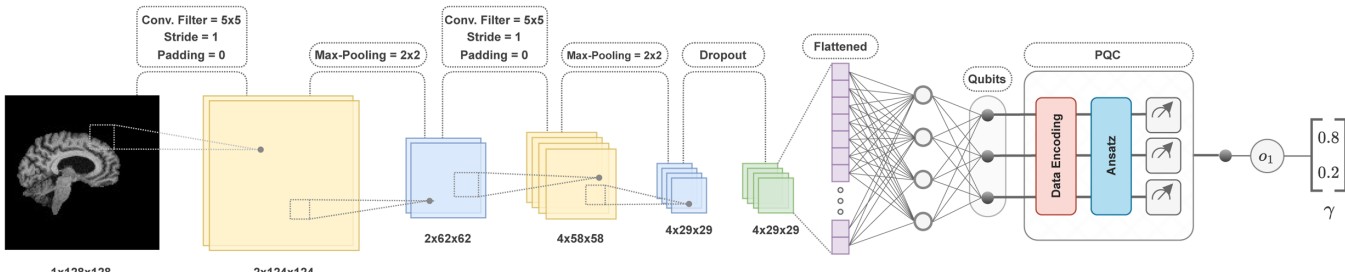

**Fig 5. The illustration depicts our CQ-CNN architecture for binary image classification.** The input is a grayscale 2D MRI slice of size 1x128x128, which passes through a convolutional layer with a 5x5 filter, a stride of 1, and no padding, producing 2x124x124 feature maps, followed by 2x2 max-pooling, which reduces it to 2x62x62. A second convolutional layer with the same filter settings generates 4x58x58 feature maps, which are then reduced to 4x29x29 through max-pooling. A dropout layer is applied for regularization, and the output is flattened for the fully connected (dense) layer. The processed data is then fed into the PQC, where classical data is encoded into quantum states, followed by ansatz layers with learnable parameters updated using the gradient descent algorithm defined in Eq 8, and finally measured to produce classification probabilities, resulting in the output vector $\gamma$.

fully connected layer with a reduced number of neurons. The output is then fed into the PQC, where classical data is encoded into quantum states, processed through quantum operations, followed by measurement and classical optimization. The measured output is then passed through a final one-dimensional classical linear layer to produce the classification probability $\gamma$. The CQ-CNN model contains only around 13.7K trainable parameters (as shown in Table 1), which is significantly lower than those of modern classical CNN models such as ResNet and DenseNet [30,31]. This low parameter count is intentional, enabling us to better evaluate the capabilities of the PQC, assess the feasibility of achieving quantum advantage, and identify potential challenges associated with its integration.

## Experiments

### Dataset

We conduct experiments with our CQ-CNN model on the OASIS-2 dataset, which contains T1-weighted 3D MRI volumes from 150 subjects [32]. The dataset is categorized into four classes: non-dementia, very mild dementia, mild dementia, and moderate dementia. However, the class distribution is highly imbalanced. Common approaches to address this issue

**Table 1. Layer-by-layer configuration and parameter count of the proposed CQ-CNN model, where $\omega$ represents the number of qubits in the PQC, and $\mathcal{Q}$ indicates the number of trainable parameters within the PQC ansatz.**

| Layer | Type | Filters/Units | Kernel Size | Stride | Padding | Output Shape | Parameters |
|---|---|---|---|---|---|---|---|
| Input | - | - | - | - | - | $1 \times 128 \times 128$ | - |
| Conv1 | Conv2D | 2 | $5 \times 5$ | 1 | 0 | $2 \times 124 \times 124$ | 52 |
| MaxPool1 | MaxPooling2D | - | $2 \times 2$ | 2 | 0 | $2 \times 62 \times 62$ | 0 |
| Conv2 | Conv2D | 4 | $5 \times 5$ | 1 | 0 | $4 \times 58 \times 58$ | 204 |
| MaxPool2 | MaxPooling2D | - | $2 \times 2$ | 2 | 0 | $4 \times 29 \times 29$ | 0 |
| Dropout | Dropout2D | - | - | - | - | $4 \times 29 \times 29$ | 0 |
| Flatten | Flatten | - | - | - | - | 3364 | 0 |
| Dense1 | Fully Connected | 4 | - | - | - | $1 \times 4$ | 13,460 |
| Dense2 | Fully Connected | $\omega$ | - | - | - | $1 \times \omega$ | $\omega_1 = (4 \times \omega) + \omega$ |
| PQC | TorchConnector | 1 | - | - | - | $1 \times 1$ | $\mathcal{Q}$ |
| Output | Fully Connected | 1 | - | - | - | $1 \times 1$ | 2 |
| Total number of trainable parameters | | | | | | | $13,718 + \omega_1 + \mathcal{Q}$ |

include oversampling techniques such as data augmentation (e.g., rotation, angle variation, exposure adjustment, zooming) or synthetic data generation using GANs or Diffusion Models [33,34]. In our study, traditional augmentation techniques are unsuitable because preserving the spatial orientation of MRI scans is critical to ensuring the model's generalizability in real-world scenarios. We also avoid using GANs due to their high data requirements, which cannot be met by our minority classes. Consequently, we resort to training Diffusion Models as our oversampling strategy to mitigate class imbalance.

MRI scans often include the skull and surrounding tissue, which contain no relevant information for AD classification. Therefore, as part of our preprocessing pipeline, we create two variants of the OASIS-2 dataset: one with skull-stripped (segmented) images (Fig 1(c), bottom row) and one without segmentation (Fig 1(c), top row). For skull stripping, we train a U-Net model on the NFBS dataset [35]. Finally, since our CQ-CNN model architecture is specifically designed for binary classification, we exclude the very mild dementia and mild dementia samples. Our experiments focus solely on distinguishing between the non-dementia and moderate dementia classes.

## Preprocessing

We begin our experiments by preprocessing the datasets, starting with the conversion of 3D MRI volumes into 2D slices using our custom 3D-to-2D conversion framework. For each 3D volume in the NFBS dataset, we extract 15 slices from both the axial and coronal planes, and 20 slices from the sagittal plane. For the OASIS-2 dataset, we extract 66 slices from the axial plane, 56 from the coronal plane, and 48 from the sagittal plane.

The NFBS dataset contains 125 MRI scans. This results in $125 \times 15 = 1,875$ slices per plane for both the axial and coronal planes, and $125 \times 20 = 2,500$ slices for the sagittal plane, totaling 6,250 2D images. To construct the test set, 105 slices are randomly selected from each of the axial and coronal planes, and 140 slices from the sagittal plane, yielding a total of 350 test images. The remaining slices are allocated to the training set, which consists of 1,770 axial and coronal images per plane and 2,360 sagittal images, totaling 5,900 training images. Corresponding brain masks are placed in the respective training and test set directories, and our segmentation model is trained accordingly. The training configuration is presented in Table 2, and the training process is illustrated in Fig 6.

The OASIS-2 dataset undergoes similar preprocessing steps and is then split into training and test sets using a 90:10 ratio. To address class imbalance, we use images from the minority class and train three separate diffusion models, one for each anatomical plane. The training configuration is also provided in Table 2, and the training progress is shown in Fig 7. Once trained, the diffusion models are used to generate synthetic images to balance the class distribution, resulting in three separate subsets of the OASIS-2 dataset, one for each plane (axial, coronal, and sagittal). These three subsets are then combined to form a multi-plane (3-plane)

**Table 2. Training configurations for the segmentation ($\mathcal{S}_{seg}$), diffusion ($\mathcal{D}_{diff}$), and classification ($\mathcal{C}_{cls}$) models.**

|  | $\mathcal{S}_{seg}$ | $\mathcal{D}_{diff}$ | $\mathcal{C}_{cls}$ |
|---|---|---|---|
| Image Size | 128 | 128 | 128 |
| Batch Size | 8 | 16 | 32 |
| Learning Rate | 0.005 | 1e-4 | 0.001 |
| Mixed Precision | – | fp16 | – |
| Optimizer | Adam | AdamW | Adam |
| Loss Function | Dice + BCE | MSE | CrossEntropy |

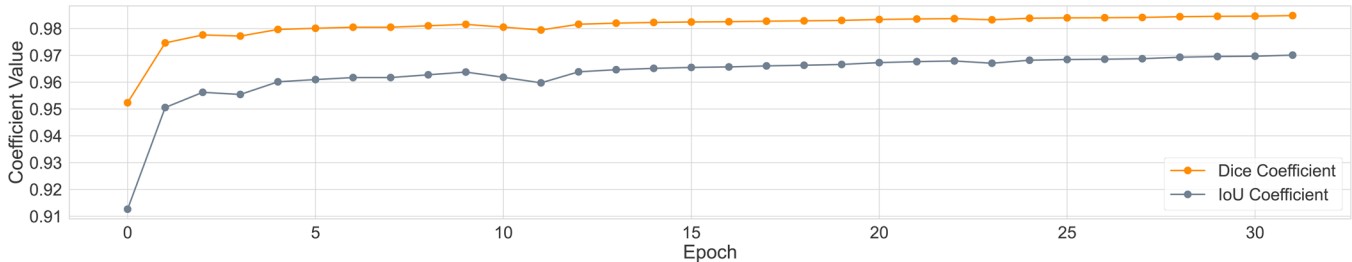

**Fig 6. The graph depicts the training progress of the segmentation model, showing the Dice and IoU coefficients over 30 epochs.** The Dice coefficient (orange) increases rapidly and stabilizes around 0.985, while the IoU coefficient (gray) converges to around 0.97.

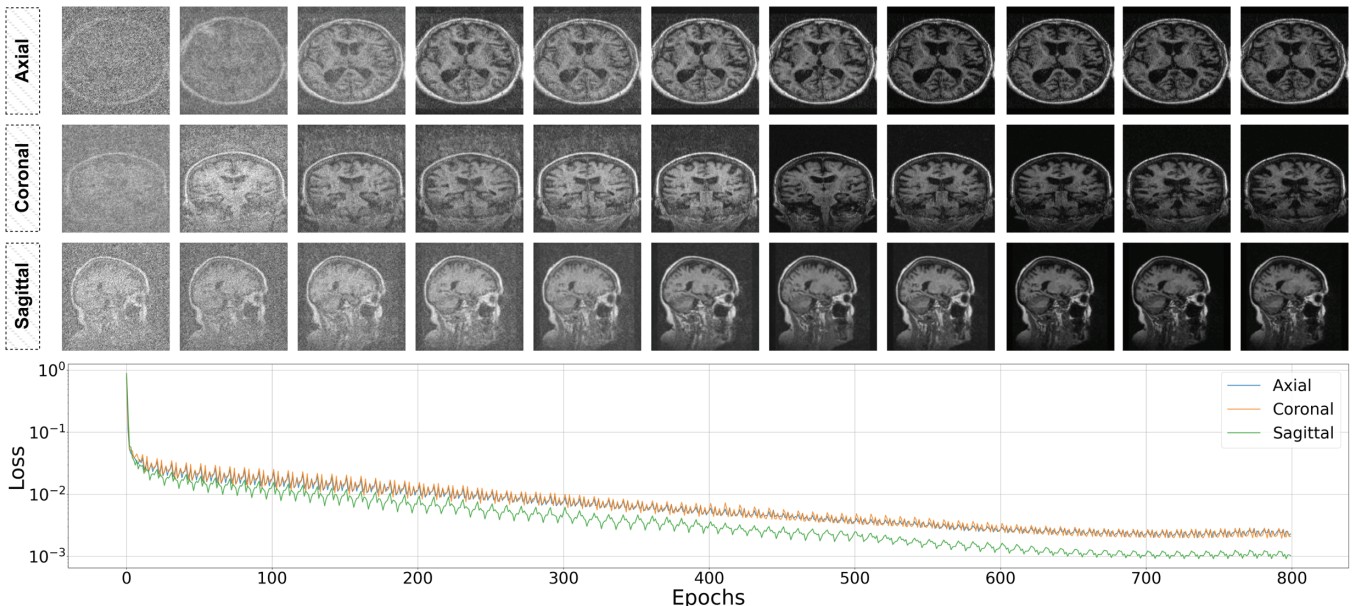

**Fig 7. The visuals present the training loss curves over 800 epochs for three distinct diffusion models.** The upper section displays the progression of generated images at different stages of training, showcasing the refinement of details as training advances. The lower graph presents the training loss curves for the three models. The y-axis, shown on a logarithmic scale, highlights the sharp decline in loss during the early stages of training. All three models follow a similar convergence pattern, with losses stabilizing around 700 epochs.

dataset. As a result, we obtain four distinct datasets, and Fig 8 presents the distribution of training and test images across them. In the final step, we apply our trained segmentation model to each of these datasets to produce skull-stripped versions of the MRI slices, resulting in four additional dataset variations containing segmented 2D images.

## Classification model training

The CQ-CNN models are trained on eight distinct subsets of the preprocessed OASIS-2 dataset. The PQCs in our CQ-CNN models are designed to execute their computations on physical quantum hardware. However, access to actual quantum hardware is currently limited, especially for research purposes, due to the scarcity and high cost of such devices [20]. Therefore, we simulate the quantum computations on classical computers using Qiskit [36]. Our experiments are conducted using 2-qubit and 3-qubit circuits. To establish a fair performance baseline, we also train purely classical models with the same number of parameters as

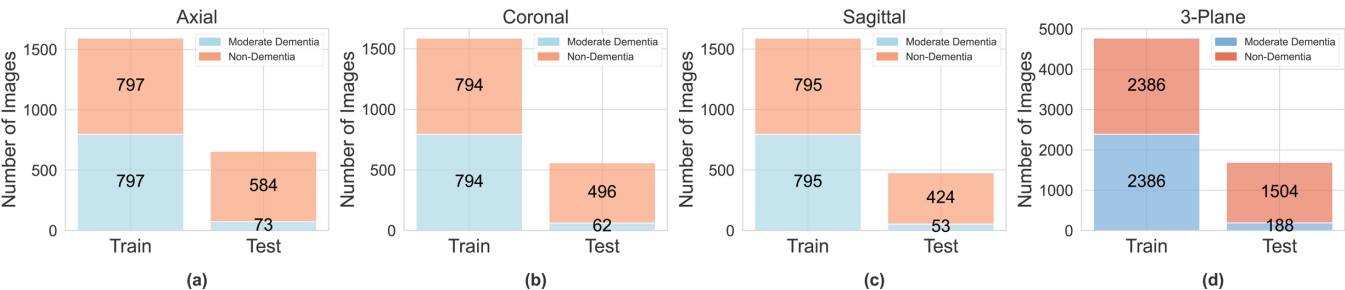

**Fig 8. Distribution of training and testing images for the moderate dementia and non-dementia classes in the OASIS-2 dataset, shown across four variations with images from different planes: (a) axial, (b) coronal, (c) sagittal, and (d) 3-plane (a combined set containing samples from all three individual planes).**

our CQ-CNN models and compare their training convergence behaviors. The configuration used for CQ-CNN training is summarized in Table 2.

## Results

### Performance analysis

The classification performance of the CQ-CNN models is summarized in Table 3, from which we observe the following.

**Effect of skull-stripping:** Models trained on skull-stripped datasets generally achieve lower scores across evaluation metrics compared to those trained on non-skull-stripped datasets. For example, the $\alpha_6$ model achieves an F1 score of 0.8088, whereas the $\alpha_5$ model

**Table 3**. **Performance analysis of CQ-CNN models across axial, coronal, sagittal, and combined 3-plane views.** Key evaluation metrics, including precision, F1-score, specificity, accuracy, and training time, are provided for models using both 2-qubit ($\alpha_i$) and 3-qubit ($\beta_i$) configurations, where $i$ represents experiments conducted on a specific dataset variation. Each metric is reported as the mean and standard deviation over multiple runs. The analysis also examines the impact of skull-stripping (denoted by $\Xi$) on model performance and compares results based on whether the models were trained with single-plane (2D) or multi-plane (3D) images. Boldface numbers indicate the best performance. The symbol ↑ denotes that a higher value is better, while ↓ signifies that a lower value is better.

| | Plane | $\Xi$ | Dim. | Qubits | Precision [↑] | F1-Score [↑] | Specificity [↑] | Accuracy [↑] | hh:mm:ss ± mm:ss Training Time [↓] |
|---|---|---|---|---|---|---|---|---|---|
| $\alpha_1$ | Axial | ✗ | 2D | 2 | 0.8777 ± 0.0492 | 0.9345 ± 0.0279 | 0.9843 ± 0.0052 | 0.9840 ± 0.0075 | 00:08:20 ± 00:17 |
| $\alpha_2$ | Axial | ✓ | 2D | 2 | **0.8955 ± 0.1235** | **0.9342 ± 0.0569** | **0.9853 ± 0.0180** | **0.9851 ± 0.0136** | 00:07:17 ± 00:43 |
| $\alpha_3$ | Coronal | ✗ | 2D | 2 | **0.9471 ± 0.0307** | **0.9727 ± 0.0162** | **0.9930 ± 0.0043** | **0.9937 ± 0.0038** | 00:04:24 ± 00:35 |
| $\alpha_4$ | Coronal | ✓ | 2D | 2 | 0.6931 ± 0.1474 | **0.8143 ± 0.1032** | **0.9405 ± 0.0395** | **0.9471 ± 0.0351** | 00:04:14 ± 00:50 |
| $\alpha_5$ | Sagittal | ✗ | 2D | 2 | **0.9569 ± 0.0610** | **0.9775 ± 0.0318** | **0.9941 ± 0.0083** | **0.9948 ± 0.0074** | 00:04:28 ± 00:44 |
| $\alpha_6$ | Sagittal | ✓ | 2D | 2 | 0.7068 ± 0.2709 | 0.8088 ± 0.1947 | 0.9404 ± 0.0642 | 0.9370 ± 0.0713 | 00:04:51 ± 00:21 |
| $\alpha_7$ | 3-Plane | ✗ | 3D | 2 | **0.9034 ± 0.1129** | 0.9246 ± 0.0489 | 0.9861 ± 0.0169 | 0.9823 ± 0.0125 | 00:25:18 ± 02:12 |
| $\alpha_8$ | 3-Plane | ✓ | 3D | 2 | 0.6721 ± 0.1411 | 0.7527 ± 0.2505 | 0.9570 ± 0.0136 | 0.9350 ± 0.0390 | 00:24:57 ± 04:06 |
| $\beta_1$ | Axial | ✗ | 2D | 3 | **0.9069 ± 0.0080** | **0.9512 ± 0.0043** | **0.9872 ± 0.0012** | **0.9886 ± 0.0011** | 00:22:40 ± 03:10 |
| $\beta_2$ | Axial | ✓ | 2D | 3 | 0.7883 ± 0.0945 | 0.8740 ± 0.0677 | 0.9686 ± 0.0167 | 0.9701 ± 0.0175 | 00:18:43 ± 01:13 |
| $\beta_3$ | Coronal | ✗ | 2D | 3 | 0.9186 ± 0.0096 | 0.9575 ± 0.0052 | 0.9895 ± 0.0022 | 0.9896 ± 0.0005 | 00:10:52 ± 01:41 |
| $\beta_4$ | Coronal | ✓ | 2D | 3 | **0.7196 ± 0.3502** | 0.8123 ± 0.2418 | 0.9283 ± 0.0955 | 0.9362 ± 0.0849 | 00:10:11 ± 01:00 |
| $\beta_5$ | Sagittal | ✗ | 2D | 3 | 0.9492 ± 0.0719 | 0.9094 ± 0.0524 | 0.9929 ± 0.0100 | 0.9811 ± 0.0089 | 00:10:15 ± 00:52 |
| $\beta_6$ | Sagittal | ✓ | 2D | 3 | **0.7814 ± 0.2334** | **0.8676 ± 0.1484** | **0.9575 ± 0.0501** | **0.9622 ± 0.0445** | 00:10:07 ± 00:20 |
| $\beta_7$ | 3-Plane | ✗ | 3D | 3 | 0.9023 ± 0.0337 | **0.9485 ± 0.0186** | **0.9864 ± 0.0052** | **0.9879 ± 0.0046** | 01:23:05 ± 27:41 |
| $\beta_8$ | 3-Plane | ✓ | 3D | 3 | **0.8319 ± 0.0686** | **0.8945 ± 0.0257** | **0.9755 ± 0.0126** | **0.9750 ± 0.0076** | 01:20:55 ± 24:41 |

attains a significantly higher score of 0.9775. However, despite their lower numerical performance, skull-stripped models provide more clinically reliable predictions, as their outputs are derived solely from brain tissue directly relevant to AD.

**Effect of qubits:** Unlike classical CNNs, where increasing the number of parameters typically enhances performance, quantum models do not always benefit from additional qubits. While a larger quantum system enables the model to capture more complex patterns, it also increases sensitivity to quantum noise, which can degrade performance. This is evident in the 2-qubit models $\alpha_3$ and $\alpha_5$, which achieve F1-scores of 0.9727 and 0.9775, compared to the 3-qubit models $\beta_3$ and $\beta_5$, with lower scores of 0.9575 and 0.9094. However, an opposite trend is observed in the 3-qubit models $\beta_6$ and $\beta_8$, which achieve F1-scores of 0.8676 and 0.8945, both higher than their 2-qubit counterparts, $\alpha_6$ and $\alpha_8$, which score 0.8088 and 0.7527. This suggests that, in certain cases, 3-qubit models can make use of their additional qubits more effectively to capture patterns in AD-relevant brain tissues compared to 2-qubit models.

**Trade-off between time and performance:** While increasing the number of qubits may occasionally improve performance, overall gains remain limited. This observation is detailed in the radar plots in Fig 9, where the 2-qubit $\alpha_i$ models and their corresponding 3-qubit $\beta_i$ models from Table 3 show similar area coverage. The primary difference is the significant increase in training time, as quantum models scale computationally with circuit depth. For example, training the 3-qubit model $\beta_8$ takes 1 hour and 20 minutes, nearly four times longer than the 24 minutes required for the 2-qubit model $\alpha_8$. A similar trend is observed across other 3-qubit models, where adding qubits doubles or triples the training time without providing proportional performance improvements.

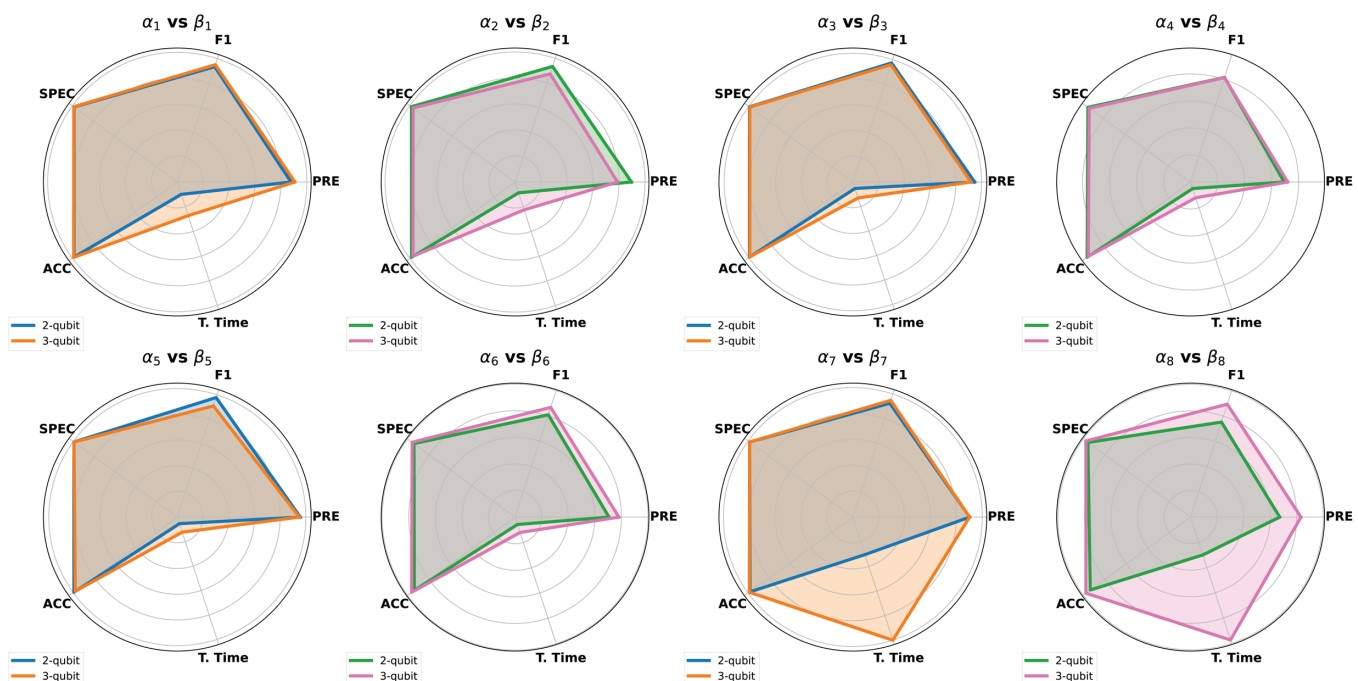

**Fig 9. Radar plots compare the performance of models with different qubit configurations across evaluation metrics: accuracy (ACC), specificity (SPEC), F1-score (F1), precision (PRE), and training time (T. Time).** Each subplot represents a comparison between the 2-qubit model ($\alpha_i$) and its corresponding 3-qubit model ($\beta_i$), where both models are trained on the same dataset $i$. The radar plots highlight that despite the use of 3-qubit models (e.g., $\alpha_7$ vs. $\beta_7$), the overall performance improvements are minimal. In contrast, training time increases significantly with the addition of qubits.

**Classical-quantum convergence analysis.** In our experiments with CQ-CNN models, we discovered a few repetitive patterns during training, particularly in the initial phases. The MRI images from both the non-dementia and moderate dementia classes are often highly similar, making it difficult for the model to discern subtle differences between the two. While quantum models are theoretically well-suited for handling high-dimensional data and capturing intricate patterns, they face practical limitations when dealing with subtle class distinctions. The primary issue arises from the quantum component of the architecture, which, despite its refined design, struggles with convergence in the early stages of training, as shown in the middle and bottom rows of Fig 10. In classical CNN models, we usually address this issue by increasing the number of parameters, enabling the model to better capture relevant features from the training data. However, when this approach is applied to quantum models by increasing the number of qubits, convergence failure worsens instead of improving.

One major reason for this instability is the inability of quantum gates to produce well-defined gradients. Quantum circuits, particularly those that use feature maps and ansatz, often result in poor gradient flow during optimization, especially when dealing with datasets in which images within the classes have fewer discriminative features. This can cause gradients to vanish or explode, making it difficult for the optimizer to adjust the quantum weights

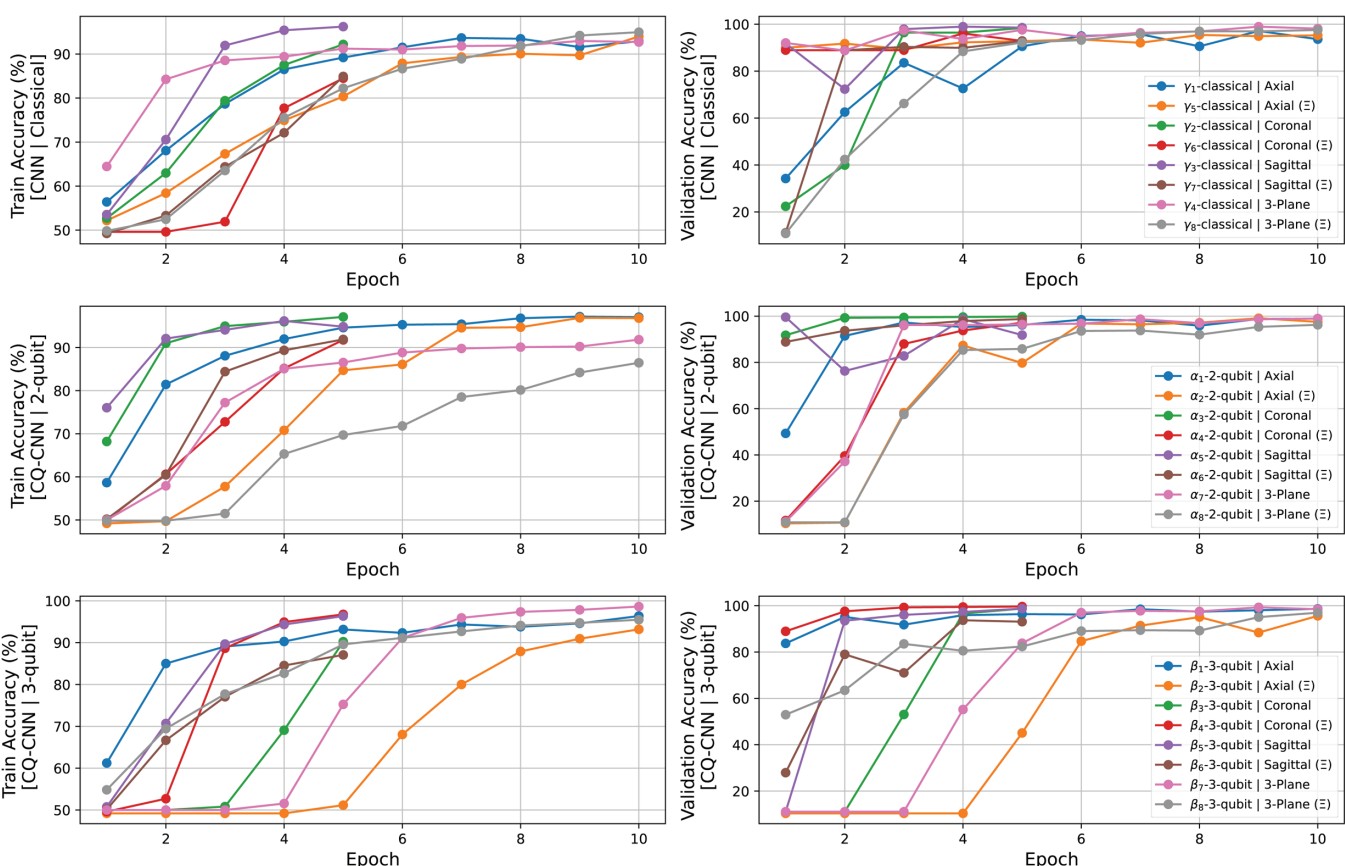

**Fig 10. The graphs present the training and validation accuracy curves for the CQ-CNN models across different MRI planes (axial, coronal, sagittal, and 3-plane) and model configurations (classical, 2-qubit, and 3-qubit), with and without skull-stripping, over several epochs.** The classical CNN (top row) shows steady, step-by-step improvement in accuracy with each epoch. In contrast, the CQ-CNN models (middle and bottom rows) exhibit slow convergence during the initial phase of training but then rapidly achieve high accuracy after a few more epochs.

effectively. The classical component, responsible for gradient-based optimization, functions well in its domain, but its optimization strategies often fail to translate smoothly to the quantum part of the model. This disconnect leads to poor convergence, particularly in the initial phase of training. As a result, CQ-CNN models often require multiple re-runs of experiments before achieving satisfactory performance.

That said, when properly converged, CQ-CNN models perform well, requiring fewer epochs than classical models to reach their potential accuracy. For example, when comparing the classical model with the 2-qubit model trained on coronal images (Fig 10, top row: classical model, middle row: quantum model, green line), the classical model requires five epochs to exceed 95% accuracy, whereas the quantum model achieves this in just two epochs. This demonstrates that the quantum advantage remains evident in our experiments, despite being overshadowed by convergence failures.

## Ablation study

**Gradient optimization algorithm tuning:** To determine which gradient optimization algorithm works best for our CQ-CNN models, we experimented with several options. Adam was the only optimizer that successfully enabled our models to converge, so we used it in all our experiments. In contrast, all other optimizers we tested, including SGD, L-BFGS, RMSprop, and Adagrad, failed to do so. Their failure can be attributed to the highly non-convex loss landscapes and gradient instability of quantum neural networks. For instance, SGD, which relies on small, incremental updates, becomes unreliable in quantum architectures due to gradient noise and non-smooth loss surfaces. L-BFGS, a second-order optimization method, assumes well-behaved loss functions, an assumption that rarely holds in hybrid quantum-classical models, leading to poor convergence. RMSprop and Adagrad, which adjust learning rates based on past gradients, struggle due to quantum parameter sensitivity, often resulting in excessively small updates that limit meaningful learning progress. In contrast, Adam's momentum-based adaptive learning strategy helps stabilize erratic gradients, making it more resilient in CQ-CNN training. Despite initial struggles, Adam eventually adapted to optimize the quantum parameters of the PQC in CQ-CNN models, enabling the model to learn effectively in the later stages of training.

**Classical parameters tuning:** We also experimented with increasing the classical parameters of the neural network by adding larger convolutional filters. However, the issue persisted, leading us to conclude that effective training of quantum models cannot be achieved simply by adding more qubits, increasing parameters, or making the architecture more complex. Instead, the focus should be on refining the gradient optimization process.

## Comparative and computational analysis

Table 4 compares our most advanced CQ-CNN models with two purely classical control models and approaches from recent studies for AD detection, focusing on both performance and computational aspects. Key insights from this comparison are discussed below.

The first notable distinction lies in the computational setup for training. Classical neural networks are typically trained on GPUs, as they benefit from mature deep learning frameworks that are optimized for GPU acceleration. In contrast, classical-quantum neural networks, including our proposed CQ-CNN models, are trained on CPUs, since there is currently no efficient mechanism to fully utilize GPU computation for such hybrid architectures. As a result, training these models on large datasets is significantly more time-consuming compared to classical models. Nevertheless, we successfully trained our CQ-CNN model on over a thousand images while maintaining rigorous experimental standards.

**Table 4. Comparison of our classical–quantum and pure classical models with recent literature approaches for AD detection, highlighting key attributes such as dataset, number of classes, model type, GPU support $\mathcal{G}^+$, segmentation usage $\mathcal{S}^+$, accuracy, parameter count, and model size.**

| Method | Year | Dataset | Class | Model Type | $\mathcal{G}^+$ | $\mathcal{S}^+$ | Accuracy | Parameters | Size (MB) |
|---|---|---|---|---|---|---|---|---|---|
| AlexNet [37] | 2020 | OASIS | 2+ | Classical | ✓ | × | 0.9285 | 60M | 227.0 |
| ResNet-50 [38] | 2021 | ADNI | 2+ | Classical | ✓ | ✓ | $0.971 \pm 0.016$ | 25.6M | 98.0 |
| DenseCNN2 [39] | 2021 | ADNI | 2 | Classical | ✓ | × | 0.9252 | 7.1M | 28.4 |
| 2D-M$^2$IC [40] | 2022 | ADNI | 2 | Classical | ✓ | × | 0.9711 | 10.3M | 39.5 |
| 3D-M$^2$IC [40] | 2022 | ADNI | 2 | Classical | ✓ | × | 0.9736 | 18.2M | 69.7 |
| Ensemble [41] | 2024 | ADNI | 2 | Classical-Quantum | ✓ | × | 0.9989 | 27.5M | 105.2 |
| ResNet-101 [42] | 2022 | ADNI/OASIS | 2 | Classical | ✓ | ✓ | 0.9078 | 44.5M | 171.0 |
| Xception [42] | 2022 | ADNI/OASIS | 2 | Classical | ✓ | ✓ | 0.8438 | 22.9M | 88.0 |
| Inception-v3 [42] | 2022 | ADNI/OASIS | 2 | Classical | ✓ | ✓ | 0.9375 | 23.9M | 92.0 |
| 2D-CNN[43] | 2024 | OASIS-3 | 2 | Classical | ✓ | ✓ | 0.7793 | 4.3M | 16.5 |
| 3D-CNN [43] | 2024 | OASIS-3 | 2 | Classical | ✓ | ✓ | 0.9167 | 5.8M | 22.3 |
| **AlexNet$^\dagger$** | **2025** | **OASIS-2** | **2** | **Classical** | ✓ | ✓ | **$0.8554 \pm 0.0610$** | **14.5K** | **0.06** |
| $\gamma_8$-classical | **2025** | **OASIS-2** | **2** | **Classical** | ✓ | ✓ | **$0.9197 \pm 0.0163$** | **13.7K** | **0.05** |
| $\alpha_8$-2-qubit | **2025** | **OASIS-2** | **2** | **Classical-Quantum** | × | ✓ | **$0.9350 \pm 0.0390$** | **13.7K + 2 qubits** | **0.05** |
| $\beta_8$-3-qubit | **2025** | **OASIS-2** | **2** | **Classical-Quantum** | × | ✓ | **$0.9750 \pm 0.0076$** | **13.7K + 3 qubits** | **0.05** |

Another key observation involves the role of brain tissue segmentation in AD classification. Many classical models, such as ResNet-50 (used by Sun *et al.* (2021) [38]), ResNet-101, Xception, and Inception-v3 (utilized by Ghaffari *et al.* (2022) [42]), as well as custom 2D and 3D CNNs (developed by Castellano *et al.* (2024) [43]), incorporate skull-stripping or brain tissue segmentation before classification. Ghaffari *et al.* used a U-Net for segmentation, while Castellano *et al.* applied the Otsu thresholding method. In alignment with these practices, we trained our own U-Net-based segmentation model to extract brain tissue from all three anatomical MRI planes. Our CQ-CNN models were subsequently trained and evaluated on both segmented and non-segmented datasets, and we showcased a direct comparison that highlights the impact of segmentation on performance. Furthermore, to promote reproducibility and facilitate future research, we have publicly released our trained segmentation model, allowing others to bypass the need for training their own segmentation networks from scratch.

Regarding classification performance and computational complexity, we highlight several key observations. Our $\beta_8$-3-qubit model achieves an accuracy of 0.9750 on the OASIS-2 dataset, which is comparable to SOTA classical models. Although the slow simulation of CPU-based quantum circuits using Qiskit limited our ability to train on larger subsets, higher-resolution images, and multi-class setups, our model still achieved high parameter efficiency without compromising performance. For instance, AlexNet [37] achieved an accuracy of 0.9285 on the OASIS dataset using approximately 60 million parameters (227 MB). In comparison, our model achieved similar performance with only 13.7K parameters (0.05 MB), which is just 0.025% of AlexNet's parameter count. To further assess how AlexNet would perform if constrained to a parameter count comparable to our CQ-CNN models, we reimplemented it as AlexNet$^\dagger$. This reduced-scale model achieved an accuracy of 0.8554, which is approximately 8.5% lower than our $\alpha_8$-2-qubit model and 12.3% lower than our $\beta_8$-2-qubit model.

Similarly, the 3D-CNN by Castellano *et al.* (2024) [43] reported an accuracy of 0.9167 on the OASIS-3 dataset using 5.8 million parameters. Our model outperformed it while using only 0.24% of that parameter count. On the ADNI dataset, models such as the ensemble by Jenber *et al.* (2024) [41] and 3D-M$^2$IC by Helaly *et al.* (2022) [40] report high accuracies of

0.9989 and 0.9736, respectively. However, these models come with significantly larger sizes (27.5 million and 18.2 million parameters) and require extensive training datasets, with up to 38,400 images at 256×256 resolution. In contrast, our model was trained on far fewer images at 128×128 resolution. Moreover, while Jenber *et al.* use a hybrid design combining classical CNN ensembles and a Quantum Support Vector Machine (QSVM) that only computes kernel values for classical classification, our model provides a fully differentiable quantum pipeline. This is made possible through the use of parameter-shift gradients that allow for direct optimization of trainable quantum parameters. In addition, several earlier methods (e.g., [40,41]) apply image augmentations such as rotations and reflections, which may distort important anatomical features in MRI scans. In contrast, our approach uses diffusion-based augmentation to generate anatomically consistent synthetic images specifically for the minority class. Taken together, these results demonstrate that our model achieves competitive performance while using significantly fewer parameters. This underscores the quantum advantage in terms of space complexity, even under computational constraints.

To better understand the contribution of the PQC within our CQ-CNN architecture, we compared the $\gamma_8$-classical model, a purely classical CNN with the same number of parameters as our $\alpha_8$-2-qubit and $\beta_8$-3-qubit models. The $\gamma_8$-classical model achieved an accuracy of 0.9197. In contrast, the $\alpha_8$-2-qubit model attained 1.63% higher accuracy, while the $\beta_8$-3-qubit model delivered a further improvement of 5.67%. These results indicate that even with an identical parameter count, the inclusion of quantum layers provides measurable performance improvements. This shows that the PQC is not redundant to the classical backbone, but instead enhances representational capacity.

## Discussion and limitations

The findings from our experiments with CQ-CNN models for AD detection can be divided into two parts.

In the first part, we investigate the potential challenges of embedding a PQC into a CNN during training. We experiment with various architectural changes, such as increasing the number of qubits, adjusting classical parameters, modifying the size of the dataset, and altering the number of classes. Through numerous trials and errors, we identify that the primary factor contributing to the model's initial low convergence is the high similarity between images from different classes.

To elaborate on this point, while our primary focus is binary classification, we also experimented with a multi-class classification setup, attempting to distinguish among four closely related AD classes. In this case, the convergence issue became significantly worse, with the model almost failing to converge. Even when it did converge, the process was extremely slow. When we reverted to binary classification, the situation improved. Interestingly, when applying a classical CNN model to the same four-class classification task, we did not encounter this problem. This suggests that the high similarity among images and the increased classification complexity negatively affect the convergence of the CQ-CNN model.

To further substantiate the claim that convergence instability arises from intrinsic data characteristics rather than model design flaws, we conducted control experiments using binary-paired subsets from the MNIST benchmark dataset, where class distributions are sufficiently distinct. As shown in Fig 11, the CQ-CNN model consistently converged with low variability, supported by non-significant ANOVA p-values, in contrast to the significant variability observed in the OASIS-2 MRI dataset. Moreover, across multiple independent runs on the MNIST control tasks, our CQ-CNN model achieved performance comparable to recent classical-quantum hybrid approaches reported by Senokosov *et al.* (2024) and Hasan *et al.*

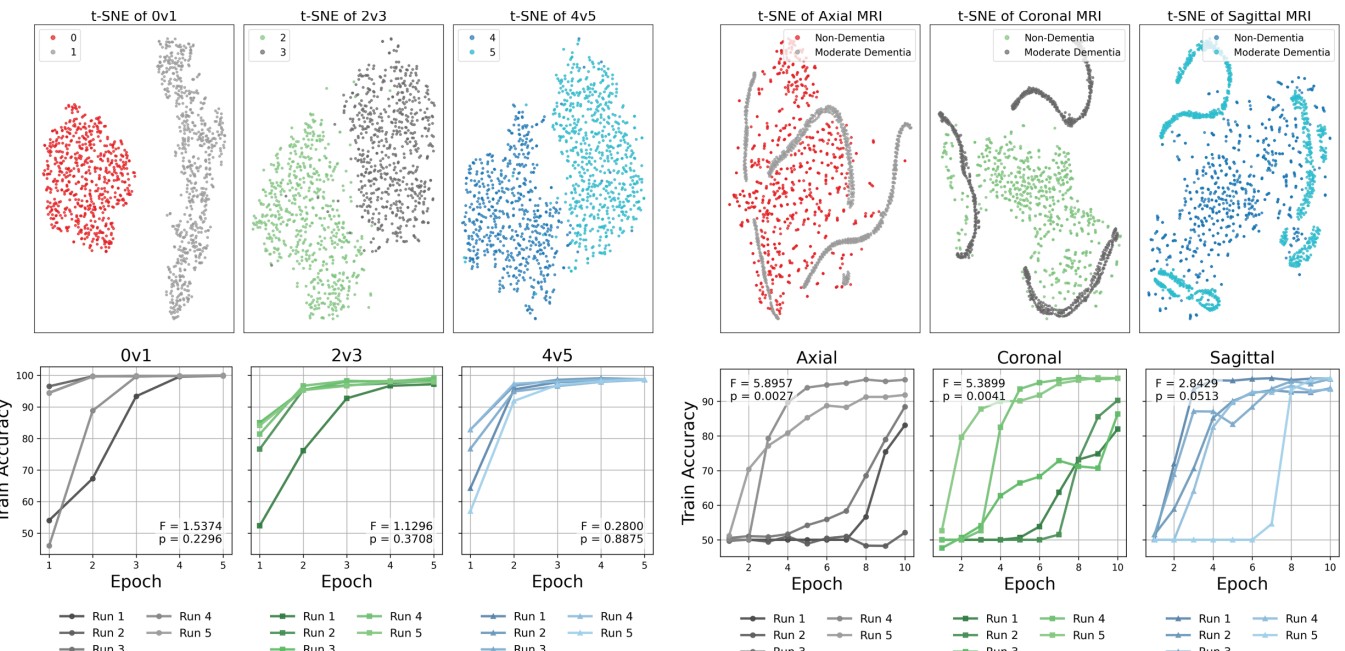

**Fig 11. Control experiments illustrate feature separability and training stability between MNIST binary pairs (0v1, 2v3, 4v5) and the OASIS-2 MRI dataset for AD classification across axial, coronal, and sagittal views.** The top row presents t-SNE visualizations of learned features, where MNIST control tasks yield well-separated clusters, while the OASIS-2 MRI views show entangled distributions between non-dementia and moderate dementia cases. The bottom row plots training accuracy across five independent runs, annotated with ANOVA F-statistics and p-values to assess variability. Consistently low variability and stable convergence in MNIST (non-significant p-values) contrast with significant variability in the axial (p = 0.0027) and coronal (p = 0.0041) views, while the sagittal view remains marginal (p = 0.0513).

(2023) [27,28]. These results collectively affirm that the convergence challenges observed in the OASIS-2 MRI data stem primarily from inherent inter-class similarity and complex feature overlap, rather than limitations of the CQ-CNN architecture itself. This data-dependent instability can also be interpreted in the context of the barren plateau phenomenon, where gradients vanish exponentially with the number of qubits or the expressibility of the quantum circuit, making optimization extremely difficult [44,45]. In our case, the quantum feature map may produce highly entangled states when processing OASIS-2 MRI data. Such entanglement could amplify gradient decay and increase the likelihood of local minima, especially when the model attempts to separate overlapping features in high-dimensional space. Conversely, the MNIST control tasks, which have simpler and more separable feature distributions, may avoid this setting, resulting in stable convergence. This theoretical framing suggests that barren plateau–like effects are more probable in datasets with subtle inter-class variations and complex quantum embeddings.

In the second part, we investigate the potential causes of low convergence in the PQC. We find that the underlying issue stems from the gradient, which is responsible for updating the model's quantum weights. If the gradient becomes trapped in a local optimum, it falsely signals that the optimal solution has been reached, preventing further improvements. As a result, the quantum weights remain unchanged, leading to poor convergence. This suggests that similar convergence issues could arise in other medical imaging classification tasks using hybrid classical-quantum architectures, though additional experiments are necessary for a definitive conclusion.

Building on these findings, we propose exploring advanced quantum optimization strategies, including layer-wise adaptive learning rates, gradient clipping, and enhanced parameter-shift rule variants, as promising approaches to stabilize gradient behavior in PQCs. Furthermore, incorporating noise-aware gradient regularization and developing more effective parameter initialization methods for quantum circuits may help navigate the non-convex and noisy loss surfaces of quantum neural networks. Moreover, recent quantum kernel learning methods introduced by Wang et al. (2025), which include a self-adaptive quantum kernel PCA for efficient dimensionality reduction and a quantum kernel-aligned regressor for modeling small, high-dimensional datasets, could be investigated to assess their potential in improving gradient optimization and overall model performance [46,47].

**Limitations:** Several limitations of our study should be acknowledged. First, our experiments were limited to configurations using only 2 and 3 qubits. Although we explored increasing the number of qubits to improve model performance, this approach proved counterproductive. This outcome was expected, as simulating quantum circuits on classical hardware becomes exponentially more demanding with each additional qubit [48]. Our comparison between 2-qubit and 3-qubit configurations (Fig 9) shows that increasing the number of qubits does not consistently enhance performance and may sometimes even degrade it. Moreover, even when performance improvements occur, the significantly longer training times required for larger qubit models often outweigh any gains. These challenges are not unique to simulations but are also present in actual quantum hardware, where increasing the qubit count amplifies the effects of quantum noise, such as decoherence, gate errors, and readout inaccuracies, all of which can degrade model accuracy. Finally, the limited accessibility and high operational costs of noisy intermediate-scale quantum (NISQ) hardware constrained our ability to evaluate the proposed models on real quantum devices [20].

## Conclusion

The automatic detection of AD is a growing research area that requires interdisciplinary expertise. A common approach to building automatic AD detection systems involves training machine learning models, such as CNNs, using 2D MRI images. Since MRI data are typically stored as 3D volumes, specialized preprocessing tools are needed to convert them into 2D slices before they can be used as training data for CNNs. However, these tools often require domain-level expertise and have a steep learning curve. Current-generation CNNs are specifically designed to run on classical hardware, making automated systems based on this architecture dependent on classical computing. With the advent of quantum computing, which aims to complement or potentially replace classical systems, it is essential to develop next-generation automated systems for AD detection that can run on quantum computers. In response to the need for multi-domain expertise and the emerging demand for quantum hardware-compatible automated systems, this paper begins by developing a simple framework to convert clinical 3D MRI volumes into 2D slices. We then propose CQ-CNN, a PQC-based lightweight hybrid classical-quantum convolutional neural network designed for binary image classification, leveraging the computational capabilities of both classical and quantum systems. Our experiments reveal a significant limitation in the current hybrid classical-quantum architecture for automated AD detection. When images of different classes in the dataset are highly similar, such as the moderate dementia and non-dementia classes in the OASIS-2 dataset, the quantum model often struggles to converge due to gradient failure. This results in minimal weight updates, causing the model to become stuck during optimization. We believe this issue may also affect other medical imaging datasets and propose it as a direction for future research. When the model does converge, it demonstrates clear signs of quantum

advantage by achieving accuracy comparable to state-of-the-art classical methods with significantly fewer parameters. For example, our $\beta_8$-3-qubit reached 0.9750 accuracy using only 13.7K parameters (0.05 MB). Overall, these findings highlight the need for further improvements in quantum optimization techniques to make current-generation hybrid classical-quantum models practical for real-world medical imaging applications such as automatic AD detection.

## Author contributions

**Conceptualization:** Mominul Islam, Mohammad Junayed Hasan, M.R.C. Mahdy.

**Data curation:** Mominul Islam.

**Formal analysis:** Mominul Islam, Mohammad Junayed Hasan.

**Investigation:** Mominul Islam.

**Methodology:** Mominul Islam.

**Project administration:** Mominul Islam, Mohammad Junayed Hasan.

**Resources:** Mominul Islam.

**Software:** Mominul Islam.

**Supervision:** Mohammad Junayed Hasan, M.R.C. Mahdy.

**Validation:** Mominul Islam, Mohammad Junayed Hasan, M.R.C. Mahdy.

**Visualization:** Mominul Islam.

**Writing – original draft:** Mominul Islam, Mohammad Junayed Hasan.

**Writing – review & editing:** Mominul Islam, Mohammad Junayed Hasan, M.R.C. Mahdy.

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
