## [Decision Letter · Decision Letter 0]

2 Jul 2025

PONE-D-25-31817CQ-CNN: A lightweight hybrid classical-quantum neural network for Alzheimer’s disease detection using clinical 3D MRIPLOS ONE

Dear Dr. Mahdy,

Thank you for submitting your manuscript to PLOS ONE. After careful consideration, we feel that it has merit but does not fully meet PLOS ONE’s publication criteria as it currently stands. Therefore, we invite you to submit a revised version of the manuscript that addresses the points raised during the review process.

We look forward to receiving your revised manuscript.

Kind regards,

Zeheng Wang

Academic Editor

PLOS ONE

Journal Requirements:

**Additional Editor Comments:**

Although reviewers suggested Rejection among some Major Revisions, I think the manuscript merits reconsideration upon satisfactory revisions. Kindly see the comments enclosed. Also, please justify by yourself whether the Ref recommendations, if any, are necessary. Please cite those only if they do improve the manuscript.

Best

Zeheng

Reviewers' comments:

Reviewer's Responses to Questions

**Comments to the Author**

1. Is the manuscript technically sound, and do the data support the conclusions?

Reviewer #1: Partly

Reviewer #2: Partly

Reviewer #3: Partly

Reviewer #4: Partly

2. Has the statistical analysis been performed appropriately and rigorously? 

Reviewer #1: No

Reviewer #2: Yes

Reviewer #3: No

Reviewer #4: No

3. Have the authors made all data underlying the findings in their manuscript fully available?

Reviewer #1: No

Reviewer #2: Yes

Reviewer #3: Yes

Reviewer #4: Yes

4. Is the manuscript presented in an intelligible fashion and written in standard English?

Reviewer #1: Yes

Reviewer #2: Yes

Reviewer #3: Yes

Reviewer #4: Yes

5. Review Comments to the Author

Reviewer #1: This paper uses CQ-CNN for Alzheimer's disease recognition and achieves performance superior to SOTA models. The following are suggestions for revising this paper:

1. Keep the introduction section concise. Many irrelevant details make it overly lengthy. Focus on explaining the shortcomings of machine learning models in AD, how your strategy addresses these major deficiencies, and why these deficiencies are significant.

2. Paper contributions are typically three points. The main theme must be highlighted. Since the title and abstract of this paper focus on quantum models, classical preprocessing methods such as diffusion models and SKULLNET should not be given excessive attention, as this obscures the main focus.

3. The main theme of the main text is not prominent, and there are numerous redundant paragraphs, causing the core method to be completely overshadowed by preprocessing methods. I had to spend a significant amount of time passively accepting basic theories like diffusion models that I already knew. If you want to propose a complete system, you should give it a separate name, such as an AD detection system based on XXX, to encompass all the content. However, the title of this paper is CQ-CNN, so you only need to focus on introducing the novelty and effectiveness of this method. It is recommended to keep the paper length around 14 pages.

4. For existing methods and models, please do not draw diagrams; simply cite the references. When describing others' methods, keep the description concise and summarise them in a short paragraph.

5. The experimental model parameter configuration is rather rough. Since this paper involves too many models (including a large number of preprocessing methods, which ultimately undermine the status of the quantum method), detailed configuration parameters such as the number of convolutional kernels and layers in the convolutional neural network are not reflected. It is strongly recommended to list a more detailed table for explanation to facilitate readers' configuration and reproduction.

6. I am contemplating a core question: Are the superior experimental results you have presented due to extensive preprocessing or the quantum model? Generally, we only perform simple dimensionality reduction on the dataset before directly inputting it into the quantum machine learning model for processing, thereby reducing the impact of various preprocessing methods on the experimental results. Could the results in this paper be due to the superior classical preprocessing? Or, even without the quantum model, the results were already good enough, and adding the quantum model only resulted in a small improvement?

7. Please use the method of controlling variables to ensure fairness in the comparison. Consider this: if the parameter size of ALEXNET were also close to 13K, would its performance be higher than 0.92?

Reviewer #2: This manuscript introduces a hybrid classical–quantum convolutional neural network (CQ-CNN) for Alzheimer's disease (AD) detection using clinical 3D MRI, with a focus on lightweight architecture and synthetic data generation via diffusion models. While the study addresses a timely and potentially impactful topic, several critical issues need to be addressed before the manuscript can be considered for publication.

a. The technical novelty of the proposed CQ-CNN architecture is not sufficiently clear. Although the combination of classical CNN layers with parameterized quantum circuits (PQCs) is conceptually relevant, similar hybrid models have already been explored in recent literature, including quantum support vector machines and hybrid quantum-classical classifiers. The manuscript does not convincingly demonstrate what new insights or architectural innovations it contributes beyond prior work, especially given the relatively simple PQC configuration (2 or 3 qubits) and standard encoding (ZZFeatureMap) applied.

b. The performance comparisons lack depth. The study compares the CQ-CNN only to a few classical models with matched parameter counts, but fails to benchmark against more competitive lightweight deep learning models or state-of-the-art AD classification frameworks. Moreover, no statistical tests are provided to assess the significance of the reported improvements. The claim of quantum advantage based on reduced epoch count is weak without clearer control experiments to rule out confounding variables such as optimizer settings, dataset variance, or model initialization.

c. The experiments rely heavily on a small binary subset of the OASIS-2 dataset (non-demented vs. moderate demented), which reduces the complexity and clinical relevance of the classification task. The exclusion of mild and very mild cases, which are clinically important for early diagnosis, undermines the model's practical utility. Additionally, the segmentation model (SkullNet) and the synthetic augmentation via diffusion models are not sufficiently validated—there is no qualitative or clinical expert evaluation of the generated samples, nor any ablation study to quantify the impact of segmentation or augmentation on final classification performance.

d. The manuscript makes claims about low resource requirements and lightweight deployment, yet no real-world deployment tests (e.g., on actual quantum hardware or edge devices) are reported. Given the current limitations of NISQ-era quantum simulators, the feasibility of using such quantum circuits in real clinical settings remains speculative.

e. The manuscript does not sufficiently engage with the latest research in the field. Recent works such as those recently published—specifically, DOI: 10.1002/advs.202506213 and DOI: 10.1002/advs.202411573—highlight more advanced and targeted approaches to hybrid quantum learning and should be discussed in the context of this work.

In conclusion, while the manuscript explores a promising direction, it requires substantial revisions in terms of experimental rigor, novelty justification, literature engagement, and claims validation. I recommend major revisions.

Reviewer #3: This manuscript presents a hybrid classical–quantum convolutional neural network (CQ-CNN) designed for Alzheimer’s disease (AD) detection using clinical 3D MRI data. The model architecture is notably lightweight, utilizing just 13K parameters (0.48MB) while achieving a reported accuracy of 97.5%, surpassing state-of-the-art (SOTA) models. The pipeline includes preprocessing with a custom segmentation model (SkullNet), probabilistic diffusion models for data augmentation, and a PQC-based quantum classifier built via Qiskit. The work contributes to the emerging field of quantum machine learning (QML) for medical imaging, aiming to demonstrate quantum advantage under resource constraints. However, major revisions are needed before it is finally accepted.

1. While the manuscript claims quantum advantage, the evidence remains circumstantial and limited to simulations. No real quantum hardware results are presented, and the simulations are small-scale (2–3 qubits), limiting generalizability. Furthermore, claims of superior convergence require rigorous statistical testing (e.g., confidence intervals, ANOVA) to establish significance beyond classical baselines.

2. The exclusion of "very mild" and "mild" dementia classes significantly weakens the clinical relevance of the binary classifier. These classes are critical for early detection. The reasoning for this exclusion is inadequately justified, and the impact of this simplification is not discussed in the limitations. Consider ordinal or multi-class classification, or explicitly discuss the limitations of binary simplification.

3. The manuscript acknowledges convergence instability due to quantum gradient vanishing/exploding problems, especially with more qubits. However, there is no exploration of advanced techniques like layer-wise learning rates, gradient clipping, or parameter-shift rule variants that could address this issue. Integrate more robust quantum optimization strategies or provide ablation studies that test the effect of different PQC structures or initialization methods.

Reviewer #4: This paper introduces an end-to-end, hybrid classical-quantum framework for Alzheimer's Disease (AD) detection. The framework integrates 3D-to-2D MRI data conversion, a brain tissue segmentation model named SkullNet, a diffusion model to address class imbalance, and a lightweight hybrid classifier, the CQ-CNN. The authors report highly competitive accuracy on the OASIS-2 dataset and claim a potential quantum advantage in terms of parameter efficiency and convergence speed.

Comments:

- The paper's emphasis on parameter efficiency, citing a total of only 13K parameters, warrants a more detailed analysis. The manuscript does not provide a parameter breakdown between the classical CNN backbone and the Parameterized Quantum Circuit (PQC). Without this information, it is difficult to assess to what extent this efficiency is attributable to the quantum component versus a potentially minimalistic design of the classical network itself. Consequently, comparing the model's total parameter count to that of large-scale classical models like AlexNet (60M parameters) may overstate the unique contribution of the PQC to the model's compactness.

- The work encompasses a complete and complex pipeline. This makes the paper's primary contribution unclear. Is the core innovation the novel CQ-CNN architecture itself, or is it the entire end-to-end clinical workflow, which integrates segmentation and diffusion models with a quantum classifier for the first time in this context? Components such as using U-Net for segmentation or diffusion models for data generation are not, in themselves, new concepts. The authors should more precisely frame their novelty around the integration and application of this full stack within a hybrid quantum computing context.

- The paper claims SOTA performance by comparing the CQ-CNN to several existing models in Table 5. However, a closer look reveals that many of the compared models were evaluated on different datasets (e.g., ADNI, OASIS-3), which makes a direct performance comparison challenging. To ensure fairness and accuracy, the authors should refine their claim to be more specific, for instance, by stating that they have achieved SOTA performance "on the OASIS-2 dataset for the specified binary classification task."

- The paper admits that the CQ-CNN models suffer from convergence issues, stemming from gradient failure in the quantum component, which necessitates "multiple re-runs of experiments before achieving satisfactory performance". If a model frequently fails to converge, can its occasional rapid convergence be robustly framed as an "advantage"? The authors should discuss this trade-off between potential speed and practical instability more deeply, and quantify this instability to allow for a proper assessment of the model's experimental reproducibility and stability.

- The paper includes a valuable ablation study by comparing results with and without skull-stripping. However, the necessity of another key, complex component—the probabilistic diffusion model—is not experimentally justified. The authors argue it is superior to traditional augmentation, but this is not demonstrated. A crucial ablation study comparing the results of the diffusion model against simpler augmentation techniques (e.g., rotation, flipping) and a no-augmentation baseline would be essential to justify the inclusion of this computationally intensive step.

6. PLOS authors have the option to publish the peer review history of their article (what does this mean?). If published, this will include your full peer review and any attached files.

Reviewer #1: No

Reviewer #2: No

Reviewer #3: No

Reviewer #4: No

---

## [Author Response · Author response to Decision Letter 1]

1 Aug 2025

The response to reviewers is attached in the "Attach Files" stage under the title "Response to Reviewers.pdf".

---

## [Decision Letter · Decision Letter 1]

16 Aug 2025

PONE-D-25-31817R1CQ-CNN: A lightweight hybrid classical–quantum convolutional neural network for Alzheimer’s disease detection using 3D structural brain MRIPLOS ONE

Dear Dr. Mahdy,

Thank you for submitting your manuscript to PLOS ONE. After careful consideration, we feel that it has merit but does not fully meet PLOS ONE’s publication criteria as it currently stands. Therefore, we invite you to submit a revised version of the manuscript that addresses the points raised during the review process.

We look forward to receiving your revised manuscript.

Kind regards,

Zeheng Wang

Academic Editor

PLOS ONE

Journal Requirements:

Additional Editor Comments:

Reviewers raised some new issues, along with concerns about poorly addressed previous comments. Please revise your manuscript according to the comments below, paying particular attention to avoid directly adopting any contexts from LLM. Please DO proofread and make all expressions tight and clear. Thanks.

Reviewers' comments:

Reviewer's Responses to Questions

**Comments to the Author**

1. If the authors have adequately addressed your comments raised in a previous round of review and you feel that this manuscript is now acceptable for publication, you may indicate that here to bypass the “Comments to the Author” section, enter your conflict of interest statement in the “Confidential to Editor” section, and submit your "Accept" recommendation.

Reviewer #1: (No Response)

Reviewer #3: All comments have been addressed

Reviewer #4: (No Response)

2. Is the manuscript technically sound, and do the data support the conclusions?

Reviewer #1: No

Reviewer #3: Yes

Reviewer #4: Yes

3. Has the statistical analysis been performed appropriately and rigorously? 

Reviewer #1: No

Reviewer #3: N/A

Reviewer #4: Yes

4. Have the authors made all data underlying the findings in their manuscript fully available?

Reviewer #1: No

Reviewer #3: Yes

Reviewer #4: Yes

5. Is the manuscript presented in an intelligible fashion and written in standard English?

Reviewer #1: Yes

Reviewer #3: Yes

Reviewer #4: Yes

6. Review Comments to the Author

Reviewer #1: Most importantly, the authors failed to directly address all of my questions, especially the last two, without providing a reasonable explanation. Second, the citation standards are not corrected. The paper is too long, and the text is overly verbose. In short, it fails to achieve the concise and straightforward approach expected of an academic paper. Third, the paper uses numerous terms interchangeably, without any consistency. This is a basic error, but it is likely a common one for GPT. For example, PQC and parameterized quantum circuit appear interchangeably. What is the meaning of the abbreviation PQC? Based on the above, I believe this paper is full of flaws and redundant content. I recommend rejection.

Reviewer #3: The authors have satisfactorily modified their manuscript according to my previous criticisms. Therefore, I recommend the publication of this manuscript.

Reviewer #4: In the new manuscript, most of my previous concerns have been well addressed:

Regarding the parameter breakdown (Comment 4.1): The newly added Table 1 provides a clear and detailed breakdown of the model's parameters, and the claims regarding parameter efficiency have been appropriately toned down and contextualized.

Regarding the primary contribution (Comment 4.2): The authors have refocused the manuscript. By streamlining the Methods section and rewriting the Introduction, they have made it clear that the core contribution is the investigation of the CQ-CNN architecture on a challenging, fine-grained classification task, rather than the preprocessing pipeline itself.

Regarding the SOTA claims(Comment 4.3): The authors have moderated their claims appropriately. The discussion now presents the model's performance as "comparable" to SOTA under significant computational constraints, which is a much more defensible and nuanced position.

Regarding the convergence instability (Comment 4.4): The authors have addressed this concern by conducting new control experiments on the MNIST dataset to investigate the model's stability. The results, presented in the newly added Figure 11, show that the architecture achieves stable convergence on this dataset, in contrast to the variability observed on the OASIS-2 dataset. This provides a basis for their conclusion that the observed instability is data-dependent rather than a result of an intrinsic architectural flaw.

Regarding the justification for the diffusion model (Comment 4.5): The authors have clarified their position by de-emphasizing the role of the diffusion model in their work. They have removed its detailed description from the "Methods" section and now present it as a standard preprocessing step within the "Experiments" section, rather than a core novel component of their contribution . Given this reframing, the use of the diffusion model is positioned as an empirical choice, which does not require the same level of experimental validation as a core novelty.

However, there are two remaining issues that should be addressed:

Regarding the SOTA claims (Comment 4.3): In the revised manuscript, the authors have appropriately shifted their focus from claiming definitive state-of-the-art (SOTA) performance to demonstrating that their model achieves "comparable performance" under significant computational constraints . To fully substantiate this revised claim, the paper would be significantly strengthened by including a direct, rigorously controlled baseline. This would involve training a classical CNN with a similar parameter count (e.g., replacing the PQC with classical dense layers) under the exact same experimental conditions (same data splits, resolution, and augmentation). Such an experiment would provide the necessary fair comparison point to empirically validate the model's performance and truly assess the PQC's contribution.

Regarding the convergence instability (Comment 4.4): The new control experiment presented in Figure 11 provides a compelling empirical finding that the model's instability is data-dependent. To elevate the impact of this finding, the manuscript should now connect this observation to a deeper theoretical discussion of known challenges in Quantum Machine Learning (QML). For instance, the discussion could posit whether the observed "gradient failure" is a manifestation of barren plateaus, which may be more likely to occur when the quantum feature map encodes highly entangled states from a dataset with subtle feature differences. Integrating this theoretical analysis would provide a more profound explanation for the observed behavior and strengthen the core findings of the paper.

7. PLOS authors have the option to publish the peer review history of their article (what does this mean?). If published, this will include your full peer review and any attached files.

Reviewer #1: No

Reviewer #3: No

Reviewer #4: No

---

## [Author Response · Author response to Decision Letter 2]

19 Aug 2025

The response to reviewers, updated manuscript, and manuscript with tracked changes are attached in the "Attach Files" stage.

---

## [Editor Report · Decision Letter 2]

22 Aug 2025

CQ-CNN: A lightweight hybrid classical–quantum convolutional neural network for Alzheimer’s disease detection using 3D structural brain MRI

PONE-D-25-31817R2

Dear Dr. Mahdy,

We’re pleased to inform you that your manuscript has been judged scientifically suitable for publication and will be formally accepted for publication once it meets all outstanding technical requirements.

Kind regards,

Zeheng Wang

Academic Editor

PLOS ONE
---

## [Editor Report · Acceptance letter]

PONE-D-25-31817R2

PLOS ONE

Dear Dr. Mahdy,

I'm pleased to inform you that your manuscript has been deemed suitable for publication in PLOS ONE. Congratulations! Your manuscript is now being handed over to our production team.

Kind regards,

on behalf of

Dr. Zeheng Wang

Academic Editor

PLOS ONE